# Molecular and Functional Diversity of Crustin-Like Genes in the Shrimp *Litopenaeus vannamei*

**DOI:** 10.3390/md18070361

**Published:** 2020-07-13

**Authors:** Shihao Li, Xinjia Lv, Yang Yu, Xiaojun Zhang, Fuhua Li

**Affiliations:** 1Key Laboratory of Experimental Marine Biology, Institute of Oceanology, Chinese Academy of Sciences, Qingdao 266071, China; lishihao@qdio.ac.cn (S.L.); xjlubio@163.com (X.L.); yuyang@qdio.ac.cn (Y.Y.); xjzhang@qdio.ac.cn (X.Z.); 2Laboratory for Marine Biology and Biotechnology, Qingdao National Laboratory for Marine Science and Technology, Qingdao 266237, China; 3Center for Ocean Mega-Science, Chinese Academy of Sciences, Qingdao 266071, China; 4University of Chinese Academy of Sciences, Beijing 100049, China; 5The Innovation of Seed Design, Chinese Academy of Sciences, Wuhan 430072, China

**Keywords:** crustin, sequence diversity, immunological function, *Litopenaeus vannamei*

## Abstract

Crustins are crustacean cationic cysteine-rich antimicrobial peptides that contain one or two whey acidic protein (WAP) domain(s) at the carboxyl terminus and mainly show antimicrobial and/or proteinase inhibitory activities. Here, we performed genome and transcriptome screening and identified 34 full-length crustin-like encoding genes in *Litopenaeus vannamei*. Multiple sequence analysis of the deduced mature peptides revealed that these putative crustins included 10 type Ia, two type Ib, one type Ic, 11 type IIa, three type IIb, four type III, one type IV, one type VI, and one type VII. These putative crustins were clustered into different groups. Phylogenetic analysis, considering their domain composition, showed that different types of crustin-like genes in crustaceans might be originated from the WAP core region, along with sequence insertion, duplication, deletion, and amino acid substitution. Tissue distribution analysis suggested that most crustin-like genes were mainly detected in immune-related tissues while several crustin-like genes exhibited tissue-specific expression patterns. Quantitative PCR analysis on 15 selected crustin-like genes showed that most of them were apparently upregulated after *Vibrio parahaemolyticus* or white spot syndrome virus (WSSV) infection. One type Ib crustin-like gene, mainly expressed in the ovary, showed the highest expression levels before the gastrula stage and was hardly detected after the limb bud stage, suggesting that it was a maternal immune effector. Collectively, the present data revealed the molecular and functional diversity of crustins and their potential evolutionary routes in crustaceans.

## 1. Introduction

Antimicrobial peptides (AMPs) are key elements of the innate immune system. AMPs are widely present in all multicellular organisms and exhibit a broad spectrum of activities against bacteria, fungi, yeast, protozoa, and viruses [1]. As of March 2020, as many as 3175 AMPs have been reported in the Antimicrobial Peptide Database (http://aps.unmc.edu/AP/main.php). These AMPs show tremendous sequence diversity, which suggests that organisms employ this mechanism to adapt to microbial challenges in different environments [2]. In invertebrates, various types of AMPs are effective components in the host innate immune system to protect them against pathogen infection. In crustaceans, especially decapods, different AMP families including penaeidins, anti-lipopolysaccharide factors, crustins, stylicins, etc., have been identified and characterized [3,4,5,6]. Significantly, more and more studies reveal that sequence diversity of these AMPs, even for the same kind of AMPs, leads to great changes in their activities, which always impels researchers to isolate new AMPs.

The whey acidic protein (WAP) domain, which was originally discovered in the mammalian milk protein, contains eight conserved cysteine residues forming a four-disulfide core [7]. In vertebrates, WAP domains also exist in non-milk secretory proteins with protease inhibitory or antimicrobial activities [8,9]. Crustins are one kind of crustacean secretory cationic cysteine-rich antimicrobial peptide that contain one or two WAP domain(s) at the carboxyl terminus [6,10]. The first crustin was identified as an 11.5 kD native protein from the hemocytes of the shore crab and subsequently named as carcinin with activity against Gram-positive bacteria [11,12]. Since then, different crustin genes have been identified in shrimp, crayfish, and crabs [13,14,15,16,17,18,19,20]. 

Crustins are characterized by diverse isoforms and various biological functions. These proteins were initially classified into three types (types I, II, and III) according to the variable N-terminal region of the mature peptides [10]. Type I crustins contain a cysteine-rich domain between the signal peptide and the WAP domain; type II crustins contain a glycine-rich domain and a cysteine-rich domain at this region; type III crustins, also called single WAP domain proteins (SWDs), have a short N-terminal region that is usually rich in proline and arginine residues [5,10]. Based on the peptide length between the cysteine-rich domain and the WAP domain, type II crustins were then divided into two sub-groups, types IIa and IIb [5,21]. Subsequently, two novel types of crustins were defined as types IV and V [6]. Type IV crustins, also named as double WAP domain (DSD)-containing proteins, contain two WAP domains in the mature peptides [22]. Type V crustins, which were first found in the ant genome, contain the cysteine-rich domain and the WAP domain, and also an extra aromatic amino acid-rich region located between them [23]. However, this type of crustin has not been identified in crustaceans until now.

Like the function of some WAP domain-containing proteins in vertebrate animals, crustins mainly exhibit antibacterial, anti-fungal, and proteinase inhibitory activities. However, different types of crustins show apparently distinct biological functions. Generally, the most reported type I crustins mainly show anti-bacterial activity against Gram-positive bacteria, while type II and III crustins exhibit anti-bacterial activity against both Gram-positive and Gram-negative bacteria [6]. Besides the anti-bacterial activity, type III crustins also possess proteinase inhibitory activity [24]. Type IV crustins also have proteinase inhibitory activity and/or anti-bacterial activity, while the function of type V crustins is still not elucidated [6].

Other biological roles of crustins are also reported. In *Penaeus monodon*, the expression of *crustinPm5* is induced upon heat shock and hyperosmotic salinity stress [25], indicating their possible functions in responding to environmental stresses. CrustinPm4 and transglutaminase I could downregulate astakine-mediated hematopoiesis through interacting with 3’-UTR of astakine mRNA, which represses translation of astakine protein [26]. In *Carcinus maenas*, the immunostaining investigation of carcinin in several organs revealed that the protein may be associated with wounding, cell damage, and/or tissue regeneration [27]. In *Portunus pelagicus*, the purified crustin Pp-Cru could trigger encapsulation and phagocytosis and exhibit a growth reduction and biofilm inhibition potential on both Gram-positive and Gram-negative bacteria [28]. In *Litopenaeus vannamei*, the crustin LvCruU might serve as a downstream effector of unfolded protein response, affecting antimicrobial immune response upon endoplasmic reticulum-stress [29]. These investigations reveal that crustins play diverse roles in various biological processes, which might be based on their various types and sequence diversities. Therefore, whole genome level identification of crustin genes in crustaceans will provide important information for understanding sequence and function diversity of these genes.

Although a previous study reported that 16 crustins could be predicted from a transcriptome of *L. vannamei* [30], their features were still not characterized. In the present study, we performed genome and transcriptome screening and identified 34 full-length crustin-like encoding genes based on the reported genome and transcriptome databases of the shrimp *L. vannamei* [31]. These genes encoded all reported types of crustins except type V and also four new types or sub-types of crustins. Phylogenetic analysis reveals their possible evolution routes and expression analysis suggests their multiple immune functions in shrimp.

## 2. Results

### 2.1. Identification and Sequence Characterization of Crustin-Like Genes in *L. vannamei*

After WAP domain query and sequence analysis in the genome and transcriptome database of *L*. *vannamei*, a total of 34 genes which encode full-length open reading frames (ORFs) of crustin-like genes were identified (Table 1). Five of them have been previously characterized in the directly submitted National Center for Biotechnology Information (NCBI) database or published literatures. Fifteen of them (including three previously characterized sequences) have been annotated in our previous shrimp genome database, while most of them have not been characterized. Seventeen of them were newly identified sequences from the assembled transcriptome database. Among the 34 sequences, 33 (except LvCruIII-4) could be mapped to the genome database, whereas 10 of them were incomplete (lacking the first 2 to 5 nucleotides in their ORF) based on the current version of the shrimp genome database. The mapping information of each gene can be found in the Appendix A.

### 2.2. Identification of New Types and Sequence Characterization of Putative Crustins in *L. vannamei*

According to the amino acids and domain composition of the deduced peptides, these 34 putative crustins could be categorized into six types, including types I, II, III, IV, VI, and VII (Figure 1). Type I was then divided into three sub-types including types Ia, Ib, and Ic. The domain composition of type Ia, composed of signal peptide, cysteine (Cys)-rich region, and the WAP domain, was the same as that of previously reported type I crustin. The domain composition of types IIa, IIb, and IV were all identical to previously reported types or sub-types of crustins. Four transcripts were identified encoding type III crustins. Among them, only LvCrustin III-3 contained a Pro/Arg-rich N-terminal region before its WAP domain, whereas the other three type III crustins had a short N-terminal region without Pro/Arg-rich characteristics.

Type Ib, Ic, VI, and VII crustins were four newly identified types or sub-types. Type Ib crustin contained a longer carboxyl terminal region (more than 30 aa) when compared to other type I crustins. Type Ic crustin contained two linked Cys-rich regions before the WAP domain. Type VI crustin was composed of signal peptide, glycine (Gly)-rich region, and the WAP domain. Type VII crustin contained the signal peptide and the WAP domain, and a serine/leucine (Ser/Leu)-rich region (32.8% of the total residues in this region, Figure 2) between them. Further searching of new identified type and subtype crustin-like genes in the transcriptome database from another shrimp, *Fenneropenaeus chinensis* [32], identified two type Ib (accession numbers: MT375591 and MT375592), one type Ic (accession number: MT375593), and one type VI crustin (accession number: MT375594), which showed high sequence similarity with those in *L. vannamei* (Figure 3).

The mature peptides of the shrimp putative crustins exhibited a wide range of theoretical isoelectric point (pI) and molecular weight (Mw) values (Figure 2). The pI values of mature peptides ranged from 4.15 to 9.69 and the Mw values ranged from 5.87 to 28.29 kD. The Cys-rich regions in all identified putative crustins shared a “CX_2~3_CX_7~13_CC” formula, where C showed the cysteine residue and X_n_ showed other residues with length ranges. The WAP domains comprised eight cysteine residues complying with a “CX_5~9_CX_6~31_CX_5_CX_5~7_CCX_3~4_CX_3~6_C” formula, where the general lengths of the second, fourth, fifth, and last X were 9~12, 5, 3, and 5, respectively. Three putative crustins, including Ia-1, Ia-9 and III-1, had an atypical WAP domain, in which the second and seventh cysteine residues were mutated. In the WAP domain of crustin VII, two extra cysteine residues existed.

### 2.3. Phylogenetic Analysis of Different Types of Putative Crustins in Shrimp

The WAP domains from the 34 identified putative crustins in *L. vannamei* were clustered by the neighbor-joining method. As shown in Figure 4, different types of putative crustins were classified into different groups. Except for Ia-2, type I crustins exhibited close relationship with each other. Type III and IV crustins also showed close relationships with type I crustins. Type II crustins were all clustered into one big group, together with Ia-2, and type VI and VII crustins.

The phylogenetic analysis of the WAP domains from different species showed that WAP domains from different types of putative crustins were categorized into four branches, branch A to D (Figure 5). Branch A WAPs, which exhibited the closest phylogenetic relationship with outgroup WAP domains, were from type Ic, III, and IV crustins. Brand B and C WAP domains were from type Ia and Ib crustins. Branch D WAP domains were from type II, VI, and VII crustins.

### 2.4. Spatial and Temporal Distribution of Crustin-Like Transcripts

Tissue distribution analysis of different types of crustin-like genes showed that most of them were widely expressed in several tissues, which were mainly immune-related tissues, including the epidermis, stomach, gill, intestine, and hemocytes (Figure 6). Several crustin-like genes exhibited tissue-specific expression patterns. Among them, Ib-2 was mainly detected in ovary, IIa-8, IIa-9, and IIa-10 were stomach specific, IIb-1 was mainly in eyestalk, and III-4 exhibited hepatopancreas-specific expression pattern. Among different developmental stages, most crustin-like genes showed an increase of expression level in shrimp after hatching (from nauplius or zoea stages, Figure 7). Two crustin-like genes, Ia-1 and Ia-5, also had a temporally high expression in the embryonic stage. In particular, the crustin-like gene Ib-2 was mainly detected in the early embryonic stages before gastrula.

### 2.5. Immune Responses of Crustin-Like Transcripts on *Vibrio parahaemolyticus* and WSSV Infection

In order to know whether these crustin-like genes participated in pathogen infection, 15 genes with a relatively high expression level in their target tissues (RPKM > 100 in the most abundant tissue) were selected to perform expression analysis in shrimp after *V. parahaemolyticus* or WSSV infection. As shown in Figure 8 and Appendix A, the expression levels of all tested crustin-like genes were obviously changed after *V. parahaemolyticus* or WSSV infection. They were upregulated at different time points, mainly at 3, 12, and 24 hpi, and exhibited similar trends in shrimp after *V. parahaemolyticus* infection (Figure 8A) or after WSSV infection (Figure 8B). An exception was the crustin-like gene Ib-1, which was downregulated after *V. parahaemolyticus* infection (Figure 8A), while being up-regulated after WSSV infection (Figure 8B).

## 3. Discussion

Crustins are important immune effectors with diversity and various functions in crustaceans and some insects. Previously identified crustins are categorized into five types, of which type I to IV crustins are from crustacean species, whereas type V crustins only exist in some ant genomes [6,23]. In the present study, as many as 34 genes encoding full-length ORF of crustin-like genes, including all reported crustin types in crustaceans, as well as two new types (types VI and VII) and two new sub-types (types Ib and Ic), were identified in the shrimp, *L. vannamei*, which was much more than those in other species and greatly enriched the diversity of crustin genes. However, the present identified crustin-like genes do not contain type V crustin either. Besides multiple protein-coding genes in the genome, crustins also contain abundant polymorphic sites. In the crab, *Portunus trituberculatus*, a total of 87 SNPs and 7 indels were obtained from a 1073 bp DNA sequence encoding *PtCrusin2* [33]. Actually, many polymorphic sites were also identified from most identified crustin-like genes in the present study (data not shown), which greatly increased the polymorphism of crustin genes. Although AMPs have been always regarded as immune effectors which show wide activities against different microbes, more and more studies reveal that some AMPs also exhibit exquisite specificity against certain pathogens [34]. In addition, different AMPs could combinate together to defend against pathogen infection [35]. We guess that a large number of crustin-like encoding genes are essential for the shrimp to defend against different pathogens in the marine environment.

Classification of crustins into different types is mainly based on the variable N-terminal region of mature peptides [10]. The C-terminal region WAP is the identical and functional domain of crustins. The phylogenetic analysis based on the WAP domains of all putative crustins from *L. vannamei* revealed that members from the same type usually had a closer relationship, like type II crustins and most type I crustins. Type III and IV crustins also exhibited close relationship with type I crustins. However, type VI and VII crustins showed close relationship with type II crustins. Moreover, all putative crustins from types I, III, and IV, except LvCrustin Ic, have a short sequence length before their WAP domains. In contrast with them, crustins from types II, VI, and VII have a relatively long N-terminal region. However, LvCrustin Ia-2, which was closely related with type II crustins, was an obvious exception in the phylogenetic data. This might be due to a fast rate of amino acid substitutions in the WAP domain of LvCrustin Ia-2, which was reported to be an important and common mode making crustins exhibit functional diversification [36].

Existence of multiple crustin genes in one species provides a basis to study the evolution of these kinds of genes in crustaceans. In a recent study, an evolution analysis on the WAP domain containing proteins in crustaceans deemed that crustins were generated into two groups by acquisition of Pro/Arg-rich region (type III) or Cys-rich region (type I) before the WAP domain, respectively. Type II crustins were generated by insertion of Gly-rich region in some type I crustins [37]. The present domain composition and phylogenetic analyses supported the viewpoint that type II crustins were generated from type I crustins by Gly-rich region insertion. However, when looking through the amino acid sequences, we could find that there is a short N-terminal region located in front of the WAP domain of most crustins. Some crustins contain several Proline residues in this short region while others don’t, which is similar with that of the short N-terminal region in different type III crustins. Therefore, considering the phylogenetic analysis data, we propose that type III crustins are the ancient type which generates type I crustins.

Based on the above discussion, we proposed an evolutionary route of crustin-like genes in crustaceans (Figure 9). The ancestral WAP core region might be inserted with a short N-terminal region to generate several ancient crustins, the type III crustins. Type Ia crustins might have originated from type III crustins by insertion of a Cys-rich region at the N-terminal, while type II crustins might be generated from type Ia crustins by a subsequent insertion of Gly-rich region before the Cys-rich region. Some type Ia and IIa crustins might also be generated from type gene duplication. The type Ic crustin might be generated by insertion of two Cys-rich regions before the WAP domain. The type IV crustin might be generated by WAP domain duplication because its two WAP domains exhibited the closest relationship compared to other WAP domains, which was also proposed to explain the existence of multiple WAP domain-containing proteins in crustaceans [37]. Type Ib crustins might come from type Ia crustins through a substitution of the stop codon generating a longer C-terminal region. For example, the substitution of the stop codon of LvCruIa-1 could lead to an extension of 39 aa at the C-terminal, which would become a new member of type Ib crustins. Seeing as there is a close phylogenetic relationship, type VI and VII crustins might be generated from type II crustins by deletion of the Cys-rich region and substitution of the Gly-rich region, respectively. However, comparative studies with broad taxonomic sampling and rigorous phylogenetic analysis are needed to clarify the evolutionary relationship of crustin genes in crustaceans.

Although crustins are only reported in crustaceans and a few insect species, the WAP four-disulfide core (WFDC) proteins are ubiquitous in many kinds of vertebrate and invertebrate animals [38]. Many secretory WFDC proteins exhibit antimicrobial activities, such as the antileukoprotease in humans [8] and waprins in snakes [39]. In crustaceans, crustins are regarded as the important immune effectors which participate in the first line of host defense to combat any invaders [40]. The tissue distribution and developmental expression patterns, as well as the immune responses to *Vibrio* and WSSV infections, of the crustin-like genes in *L. vannamei* entirely supported the opinion. Crustins in other crustaceans were also responsive to *Vibrio* or WSSV infection, suggesting their important roles as a kind of AMP [6]. Identification of many novel crustin-like genes in *L. vannamei* provides a useful source of peptides for developing new antimicrobial drugs, which will be further investigated. What is noteworthy is that one crustin-like gene, LvCrustin Ib-2, is mainly distributed in the ovary (Figure 6) and expresses in the early developmental stages before gastrula (Figure 7), suggesting its possible role as a maternal immune effector. Previously, we found that many immune-related genes, including 13 crustin encoding genes, might be involved in immune protection during shrimp molting [31]. These crustins, which were mainly detected in the epidermis, stomach, and gill, showed the lowest expression levels during late premolt and increased expression levels immediately after ecdysis [31]. In the green lip abalone *Haliotis laevigata*, the Perlwapin, a secreted nacre protein with three WAP domains, could bind mineral crystals in the shell matrix and play a major role in shell formation by inhibiting calcium deposition [41]. These molting-related crustins might also play functions in calcium deposition in shrimp, which needs to be further investigated.

## 4. Materials and Methods

### 4.1. Database

The RNA-seq data from different tissues, developmental stages and molting stages of the shrimp *Litopenaeus vannamei* were downloaded from NCBI (SRA SRR1460493−SRR1460495, SRR1460504−SRR1460505, and SRX1098368−SRX1098375). To get high-quality clean reads, algorithms were run for removing empty reads, adaptor sequences, and low-quality sequences. The clean reads of each group were then assembled into Unigenes using the RNA-Seq de novo assembly program Trinity (v2.8.2, https://github.com/trinityrnaseq/trinityrnaseq/releases) with default parameters. The gene abundances were calculated and normalized to reads per kilobase per million reads (RPKM).

### 4.2. Sequence Identification

The amino acid sequences decoding WAP domain of different types of crustins (accession numbers: BAD15063, AAZ76017, ABP88042, ABV25094, ABW82154, ACL97374, ADC32522, ACT82963, ACZ43782, AAS59736, ACQ66004) were downloaded from NCBI protein database (https://www.ncbi.nlm.nih.gov/protein). The WAP domain sequences were used as query sequences to perform a tblastn search (*E*-value < 10^−5^) using the *L. vannamei* genome database (http://www.shrimpbase.net/vannamei.html) and transcriptome databases. The acquired nucleotide sequences were analyzed with ORFfinder (https://www.ncbi.nlm.nih.gov/orffinder/) to predict their open reading frames (ORFs). The deduced amino acid sequences were then analyzed by InterPro (http://www.ebi.ac.uk/interpro/) and SignalP (http://www.cbs.dtu.dk/services/SignalP/) to predict the domains and signal peptide sequences of these deduced amino acid sequences. Those with full-length ORFs, signal peptides, and one or two WAP domains, while not containing other domains, were deemed as crustin candidates. These crustin candidates were then analyzed by performing a functional annotation with blastp on NCBI (https://www.ncbi.nlm.nih.gov). The sequences which hit a reported crustin or crustin-like sequence with the *E*-value < 10^−5^ were identified as crustin-like genes in shrimp. The identified crustin-like genes were then re-mapped to the genome database at the NCBI website (https://www.ncbi.nlm.nih.gov/) to obtain the gene encoding sequences by performing a blastn search with the *E*-value < 10^−5^.

### 4.3. Sequence and Expression Analysis

The deduced mature peptides of identified putative crustins were analyzed by performing a multiple alignment using online ClustalW2 software (http://www.ebi.ac.uk/Tools/msa/clustalw2/) with manual modification. The cysteine-rich region was assured by four conserved cysteine residues. Glycine or other amino acid-rich regions were predicted by analysis of amino acid composition using an online ProtParam tool (https://web.expasy.org/protparam/). Cluster analysis was performed with MEGA5 software (https://www.megasoftware.net/) under the neighbor-joining (NJ) method using the WAP domains of the identified *L. vannamei* putative crustins. A phylogenetic tree was constructed with the same method using the WAP domains of crustins from different crustaceans and the ant, and the WAP domains of Perlwapin from the abalone *Haliotis laevigata* (accession number: P84811) and *Haliotis asinina* (accession number: P86730) as the outgroup.

As part of the identified crustin-like genes were not annotated in the shrimp genome database, the expression profiles of these crustin-like genes were analyzed using the assembled transcriptome data. The RPKM values of these crustin-like genes in different samples including tissues, developmental, and molting stages of the shrimp were achieved from the transcriptome data. The heatmaps were drawn with the online software Morpheus (https://software.broadinstitute.org/morpheus/).

### 4.4. Animals, Infection, and Tissue Collection

Healthy Pacific whiteleg shrimp with a body weight of 9.5 ± 0.4 g were collected from our laboratory (Qingdao, China). The shrimp were cultured in culture tanks filled with aerated seawater at 26 °C and fed thrice daily with artificial food pellets for three days to acclimatize to laboratory conditions.

A total of 135 shrimps were used for infection experiment. The animals were randomly divided into three groups including the PBS group (negative control), Vp group, and WSSV group, with 45 shrimps in each group. In the PBS group, 10 μL PBS was intramuscularly injected into each individual at the 5th abdominal segments. In the Vp group, 10 μL PBS containing 10^7^ CFU formalinized *Vibrio parahaemolyticus* was injected into each individual. In the WSSV group, 10 μL PBS containing 1000 copies of WSSV particles was injected into each individual. The hepatopancreas, intestine, stomach, and gill from 9 individuals from each group were sampled as three replicates at 3, 6, 12, and 24 h post WSSV injection (hpi) for RNA extraction, respectively. The hemolymph was collected from the ventral sinus located at the first abdominal segment using a syringe with an equal volume of precooled anticoagulant solution (115 glucose, 27 sodium citrate, 336 NaCl, 9 mmol L^−1^ EDTA·Na_2_·2H_2_O, pH 7.4). Then the hemocytes were harvested by centrifugation at 800 *g*, 4 °C, for 10 min and preserved in liquid nitrogen.

### 4.5. RNA Extraction and cDNA Synthesis

Total RNA from each sample was isolated with RNAiso Plus (TaKaRa, Kyoto, Japan) following the manufacturer’s instructions. The RNA quality was assessed by electrophoresis on 1% agarose gel. About 1 μg total RNA of each sample was first treated with RQ1 RNase-Free DNase (Promega, Madison, WI, USA) and then used to synthesize cDNAs by PrimeScript™ 1st strand cDNA Synthesis Kit (TaKaRa, Kyoto, Japan) with random 6 mers.

### 4.6. Quantitative PCR and Data Analysis

Fifteen crustin-like encoding genes were selected to detect their immune responses against *V. parahaemolyticus* or WSSV challenges. The qPCR primers (Table 2) of the selected nucleotide sequences were designed with Primer Premier 5.0 (Premier Biosoft, USA). The qPCR reactions were carried out using SYBR (TOYOBO, Japan) and 18S rRNA gene was used as the internal control gene. The relative gene expression levels were calculated using the comparative Ct method with the formula 2^−ΔΔCt^. All data were acquired from at least three parallel tests in separate tubes. An independent sample *t*-test was used to analyze the difference between the data from the challenge group and PBS group at each time point by SPSS 21.0. Significant differences were considered at *p* < 0.01.

### 4.7. Ethical Statement

The present study used shrimp as experimental animals, which are not endangered invertebrates. In addition, there was no genetically modified organism used in the study. According to the national regulation (Fisheries Law of the People’s Republic of China), no permission was required to collect the animals and no formal ethics approval was required for this study.

## 5. Conclusions

In conclusion, the present study identified 34 transcripts encoding full-length crustin-like genes from the shrimp *L. vannamei*. They were classified into six types based on their N-terminal regions, with two newly reported types and two new sub-types. The domain composition and phylogenetic analysis results indicated that different types of crustins in crustaceans might have originated from the WAP core region, along with sequence insertion, duplication, deletion, and amino acid substitution. The supposition provided a primary understanding of crustin evolution in crustaceans, which need more evidence to be supported. These multiple genes and their expression patterns reveal the sequence and function diversification of crustins in crustaceans. Future investigations on the antimicrobial activities and biological functions of newly reported crustins are needed to enrich the understanding of the physiological roles of crustins in crustaceans, as well as develop new antimicrobial drugs.

## Figures and Tables

**Figure 1 marinedrugs-18-00361-f001:**
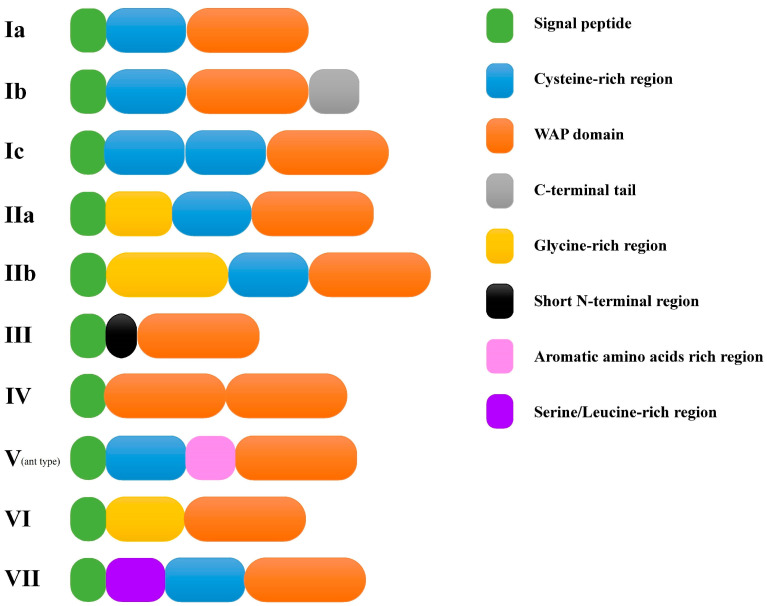
Diagram of different types of crustins. Seven different types of crustins are shown with Roman numerals from I to VII. Subtypes are displayed with lowercase letters followed the type names. Sequence features are shown with different colors.

**Figure 2 marinedrugs-18-00361-f002:**
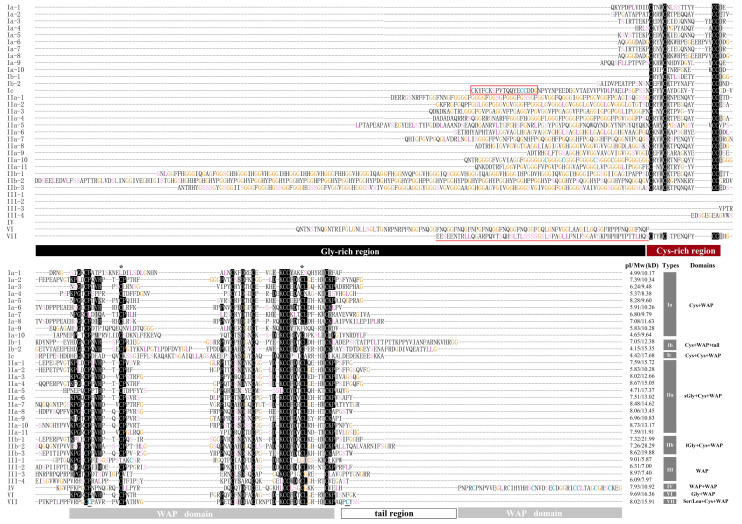
Multiple alignments of putative crustins identified in *L. vannamei*. The information of these sequences was listed in Table 1. Different sequence features including glycine-rich region, cysteine-rich region, whey acidic protein (WAP) domain, and C-terminal tail are displayed with columns under the alignment. The first cysteine-rich region of LvCrustin Ic and the serine/leucine-rich region of LvCrustin VII are boxed and underlined, respectively. Identical and similar residues are shown in dark. The second and seventh conserved cysteine sites of three atypical WAP domains are shown with stars (*) above the alignment. The cysteine, glycine, serine, and leucine residues are colored. The pI/Mw values, type, and domain composition of each putative crustin are exhibited at the end of the alignment.

**Figure 3 marinedrugs-18-00361-f003:**
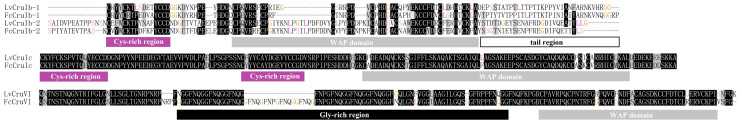
Alignment of type Ib, Ic, and VI crustins from *L. vannamei* and *F. chinensis*. Different sequence features including glycine-rich region, cysteine-rich region, WAP domain, and C-terminal tail are displayed with columns under the alignment.

**Figure 4 marinedrugs-18-00361-f004:**
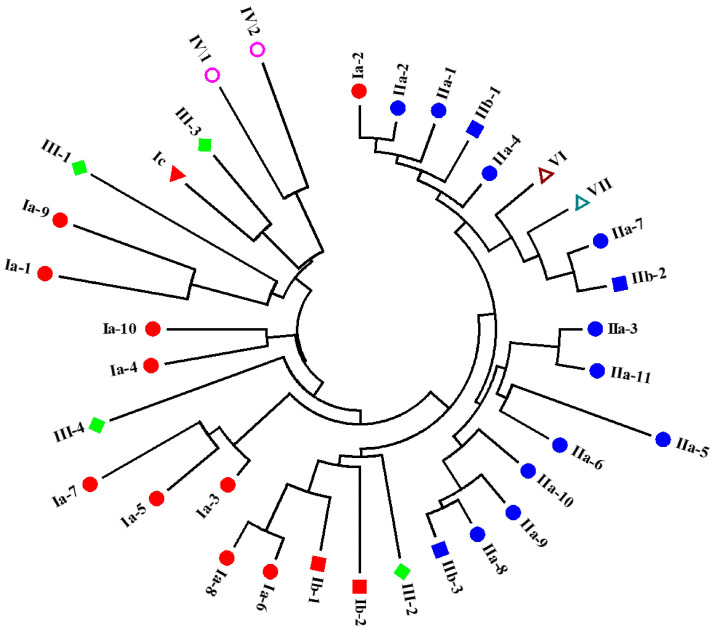
Cluster of putative crustins identified in *L. vannamei*. The WAP domains of these putative crustins are used for analysis using the neighbor-joining method. Different types and sub-types of putative crustins are shown in different shapes and colors.

**Figure 5 marinedrugs-18-00361-f005:**
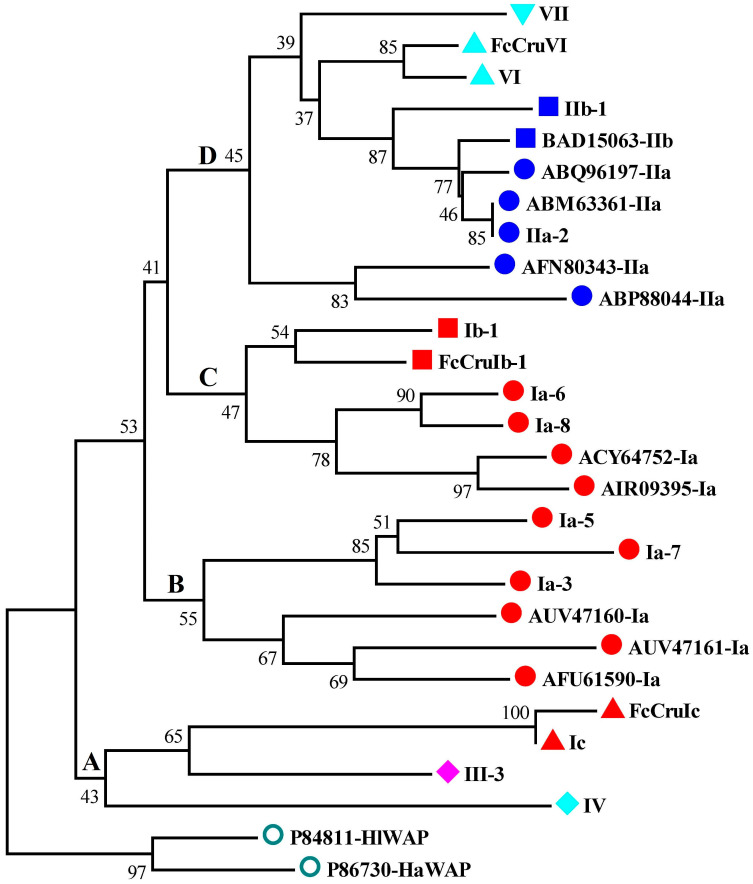
Phylogenetic analysis crustins and homologues from *L. vannamei* and other species. The WAP domains of these proteins are used for phylogenic analysis using the neighbor-joining method. Bootstrap value is set at 1000. Crustins from the present study are shown with their symbols. Crustins from other species are shown with NCBI accession numbers and types. Different types and sub-types of crustins, which are classified into four branches (**A**–**D**), and the Perlwapin are shown in different shapes and colors.

**Figure 6 marinedrugs-18-00361-f006:**
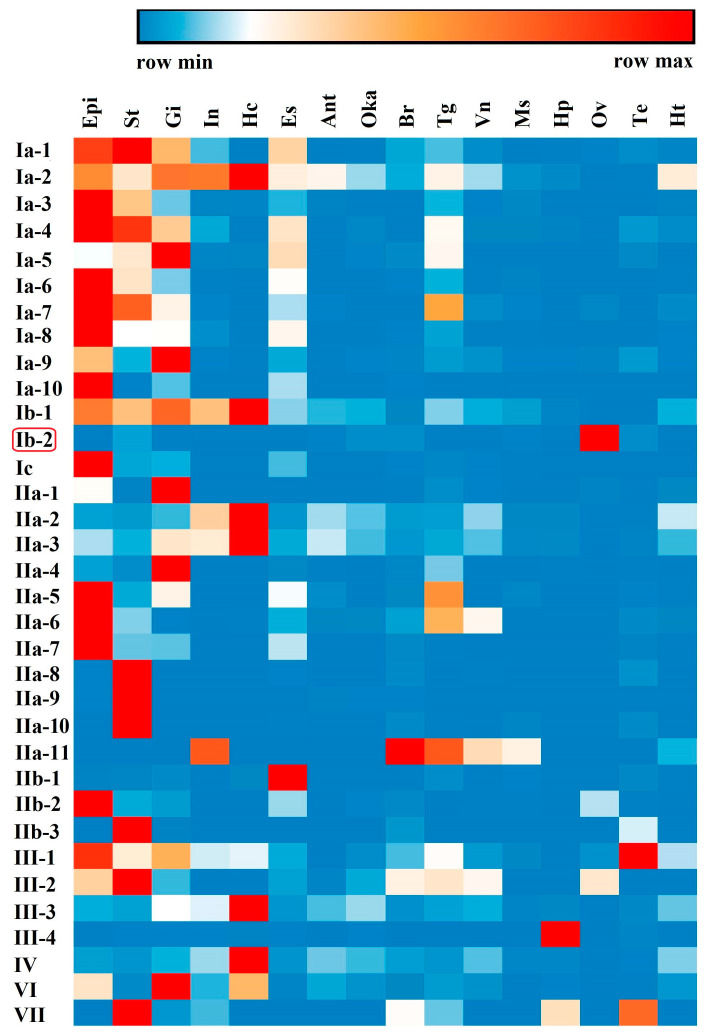
Tissue distribution of the crustin-like transcripts identified in *L. vannamei*. Epi, epidermis; St, stomach; Gi, gill; In, intestine; Hc, hemocytes; Es, eyestalk; Ant, antennal gland; Oka, lymph organ; Br, brain; Tg, thoracic ganglia; Vn, ventral nerve cord; Ms, muscle; Hp, hepatopancreas; Ov, ovary; Te, testis; Ht, heart. The gene symbols of crustin-like genes are listed left. The above color bar shows the expression level changes of the genes. The data show a mean expression level of each gene in each sample, which is a mixed tissue sample collected from 10 individuals.

**Figure 7 marinedrugs-18-00361-f007:**
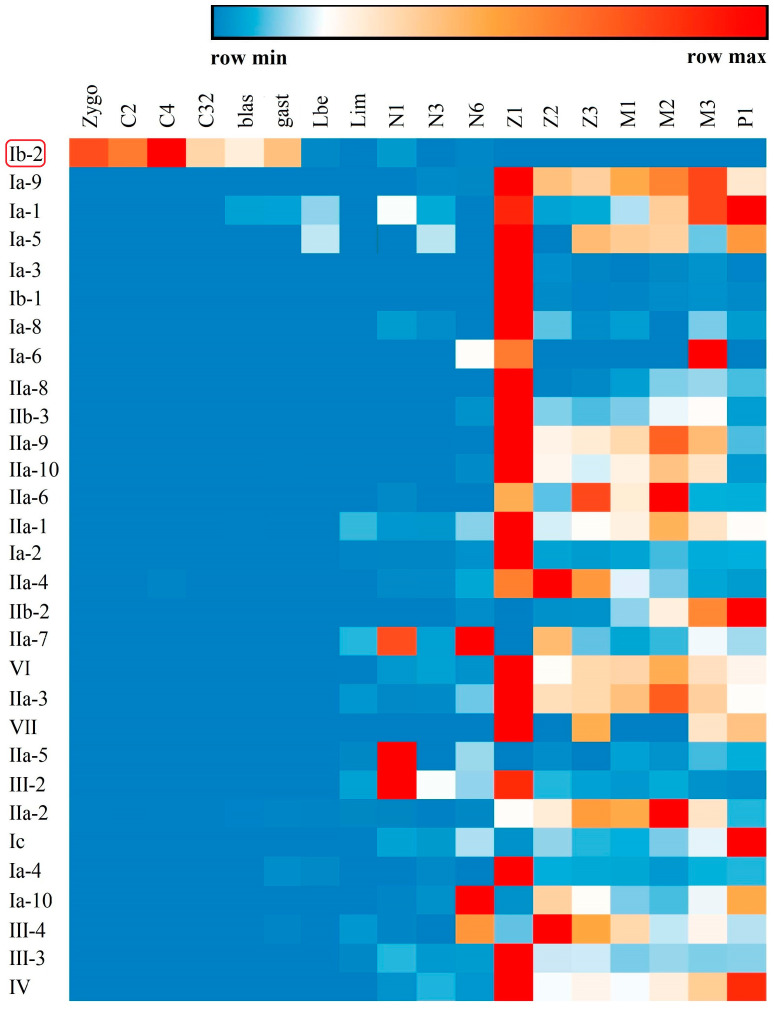
Temporal expression profiles of the crustin-like genes transcripts identified in *L. vannamei*. The expression levels of these crustin-like genes are detected in different developmental stages including fertilized egg (Zygo), 2-cell stage (C2), 4-cell stage (C4), 32-cell stage (C32), blastocyst (blas), gastrula (gast), limb bud embryo (lbe), larva in membrane (lim), nauplius I (N1), nauplius III (N3), nauplius VI (N6), zoea I (Z1), zoea II (Z2), zoea III (Z3), mysis I (M1), mysis II (M2), mysis III (M3), and postlarvae 1 (P1) stages. The gene symbols of crustin-like genes are listed left. The above color bar shows the expression level changes of the genes.

**Figure 8 marinedrugs-18-00361-f008:**
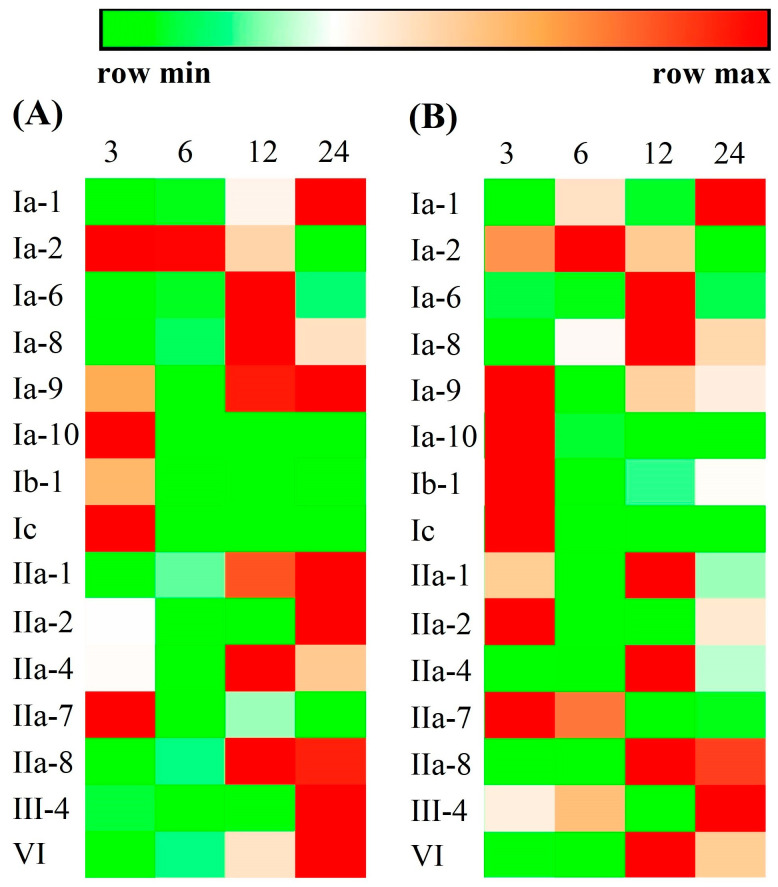
Immune responses of selected crustin-like genes against *Vibrio parahaemolyticus* (**A**) or WSSV (**B**) infection. The expression levels are analyzed in shrimp at 3, 6, 12, and 24 h post pathogen infection. The gene symbols of crustin-like genes are listed left. The above color bar shows the expression level changes of the genes. The heatmap is drawn with the mean relative expression data obtained by qPCR analysis from three biological replicates. The statistical analysis results are shown in Appendix A.

**Figure 9 marinedrugs-18-00361-f009:**
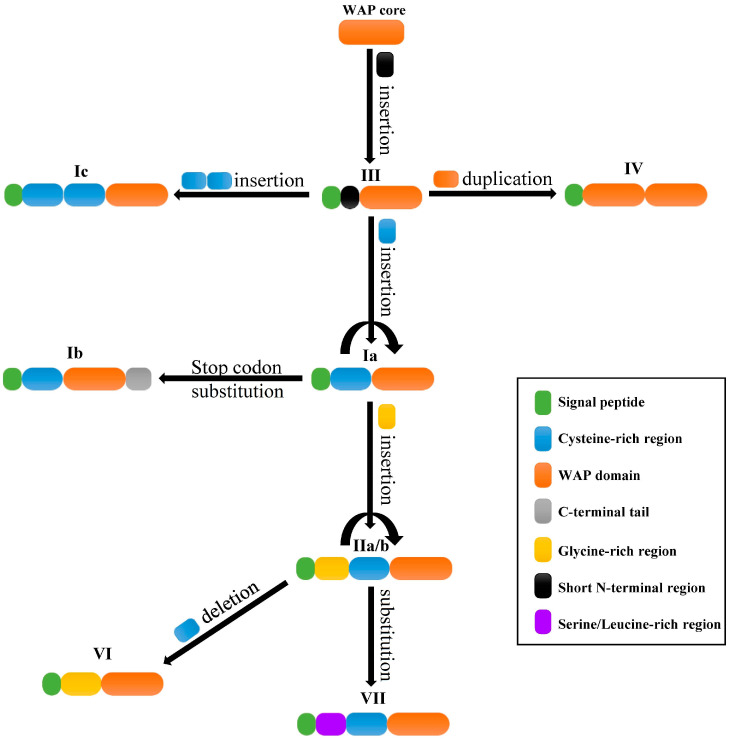
A proposed mode for the evolutionary route of crustins in crustaceans. Different types pf crustins are shown with type names and diagrams consisting of sequence features with colors. The possible changes among different types of crustins are displayed beside the arrowed lines.

**Table 1 marinedrugs-18-00361-t001:** Sequence information of crustin-like genes in *Litopenaeu vannamei.*

Transcriptome_ID	Genome_ID	Gene Symbol	NCBI Accession Number	ORF Length (bp)	Sequence Features
Unigene0010238	XP_027238377	Ia-1	MT375557	318	SP+Cys+WAP
Unigene0016512	N/A	Ia-2	MT375558	345	SP+Cys+WAP
Unigene0015168	N/A	Ia-3	MT375559	306	SP+Cys+WAP
Unigene0058358	XP_027212735	Ia-4	MT375560	291	SP+Cys+WAP
Unigene0088991	N/A	Ia-5	MT375561	309	SP+Cys+WAP
Unigene0053725	ROT80205	Ia-6	MT375562	327	SP+Cys+WAP
Unigene0088992	N/A	Ia-7	MT375563	309	SP+Cys+WAP
Unigene0053724	XP_027219667	Ia-8	MT375564	363	SP+Cys+WAP
Unigene0062421	XP_027221901	Ia-9	MT375565	342	SP+Cys+WAP
Unigene0047792	XP_027209751	Ia-10	MT375566	339	SP+Cys+WAP
Unigene0045871	XP_027212782	Ib-1	MT375567/ATU82299	381	SP+Cys+WAP+tail
Unigene0068591	XP_027219394	Ib-2	MT375568	474	SP+Cys+WAP+tail
Unigene0070992	N/A	Ic	MT375569	546	SP+Cys+Cys+WAP
Unigene0074242	N/A	IIa-1	MT375570	522	SP+sGly+Cys+WAP
Unigene0014103	XP_027208055	IIa-2	MT375571/AAL36890	492	SP+sGly+Cys+WAP
Unigene0047624	N/A	IIa-3	MT375572/AFV77524	444	SP+sGly+Cys+WAP
Unigene0075484	N/A	IIa-4	MT375573	492	SP+sGly+Cys+WAP
Unigene0059807	N/A	IIa-5	MT375574	540	SP+sGly+Cys+WAP
Unigene0070231	XP_027213162	IIa-6	MT375575	456	SP+sGly+Cys+WAP
Unigene0046286	XP_027227947	IIa-7	MT375576	531	SP+sGly+Cys+WAP
Unigene0003589	N/A	IIa-8	MT375577	471	SP+sGly+Cys+WAP
Unigene0051595	N/A	IIa-9	MT375578	384	SP+sGly+Cys+WAP
Unigene0060777	N/A	IIa-10	MT375579	465	SP+sGly+Cys+WAP
Unigene0106537	N/A	IIa-11	MT375580	405	SP+sGly+Cys+WAP
Unigene0076000	N/A	IIb-1	MT375581	732	SP+lGly+Cys+WAP
Unigene0021981	N/A	IIb-2	MT375582	870	SP+lGly+Cys+WAP
Unigene0035165	N/A	IIb-3	MT375583	720	SP+lGly+Cys+WAP
Unigene0091111	N/A	III-1	MT375584	492	SP+sN+WAP
Unigene0069142	XP_027213161	III-2	MT375585	252	SP+sN+WAP
Unigene0068231	XP_027228308	III-3	MT375586/AY465833	282	SP+sN+WAP
Unigene0047523	N/A	III-4	MT375587	282	SP+sN+WAP
Unigene0075840	N/A	IV	MT375588/ABR19819	276	SP+WAP+WAP
Unigene0073524	XP_027229424	VI	MT375589	351	SP+Gly+WAP
Unigene0106295	XP_027236662	VII	MT375590	516	SP+Ser/Leu+Cys+WAP

Note: N/A represents lack of annotated genes in the shrimp genome data. The abbreviations represent signal peptide (SP), cysteine-rich region (Cys), WAP domain (WAP), short glycine-rich region (sGly), long glycine-rich region (lGly), short N-terminal region (sN), and serine/leucine-rich region (Ser/Leu), respectively.

**Table 2 marinedrugs-18-00361-t002:** Sequence information of primers used in the present study.

Gene Symbol	Primer Name	Nucleotide Sequence (5’~3’)	Annealing Temperature (℃)
Ia-1	Ia-1F	CTCTGGTGGACATTGACTGC	55
	Ia-1R	GTTGAGGGCGTTGTGGTT	
Ia-10	Ia-10F	GGGTTGGCGTTACTTGGA	52
	Ia-10R	GGGCAGGTTGGAAACTCATT	
Ia-2	Ia-2F	CCCACAAGTCCGTCCCACCT	58
	Ia-2R	CTCCCAGACACCTGTCGAAGCAG	
Ia-6	Ia-6F	CGCCGATGGTTGCAGGTATT	55
	Ia-6R	CATGATGGTCCAAGCAGGTGTC	
Ia-8	Ia-8F	CGCCGATGGTTGCAGGTAT	55
	Ia-8R	CGTGATGGTCCAAGCAGGTGT	
Ia-9	Ia-9F	AATGGCGACCCTGATGTTG	55
	Ia-9R	GGTGTCTAAGACGTTCTGCTCC	
Ib-1	Ib-1F	TGCGATGGCGGGAAAGAT	55
	Ib-1R	GCAGCACTTCTCGTAGGGTTGG	
Ic	Ic-F	GCAGTGCCTCAATCTTATCCTC	52
	Ic-R	TCGGCTGTTACACCTTCATCTT	
IIa-1	IIa-1F	ACCGATCTTGAACCCGAGGGC	58
	IIa-1R	TGCAGGCACCTGTCGAAGCAG	
IIa-2	IIa-2F	GGTGTTGGTGGTGGTTTCCC	55
	IIa-2R	CAGTCGCTTGTGCCAGTTCC	
IIa-4	IIa-4F	TTGTGCTGTCTCCTGGTGGCTGTT	58
	IIa-4R	CGCCAAAGAATCTTGCATTCCCTC	
IIa-7	IIa-7F	GCAGGTCACCCGAGAATCAGT	55
	IIa-7R	CGAGGCATCTGTCGAAGCAG	
IIa-8	IIa-8F	GGCCTCATAGCGGGCATCG	58
	IIa-8R	CCTGGTTCTGGGGCGTCTTG	
III-4	III-4F	GGAAATCCAGTCCGGCACTCT	56
	III-4R	AAGCAGCGGTCCCAGCAA	
VI	VI-F	CCCAGAACACCAACAGCACC	55
	VI-R	GTCCACCGACGAAGTTACCG	
18S rRNA	18S-F	TATACGCTAGTGGAGCTGGAA	55
	18S-R	GGGGAGGTAGTGACGAAAAAT	

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
