# Peer review of "Molecular and Functional Diversity of Crustin-Like Genes in the Shrimp Litopenaeus vannamei"

_marinedrugs, 2020, doi:10.3390/md18070361_

Round 1

Reviewer 1 Report

Manuscript Number: marinedrugs-799897

 The manuscript entitled «Molecular and functional diversity of crustin genes provides insights into their evolution in the shrimp Litopenaeus vanname» by Shihao Li, Xinjia Lv, Yang Yu, Xiaojun Zhang  and Fuhua Li presents research aimed at screening of the shrimp Litopenaeus vannamei genome and transcriptome in order to find novel genes of crustin antimicrobial peptides (AMPs). As a result, 34 crustin encoding genes were identified. Among them, there are peptides, which comprise two novel crustin types VI and VII, as well as novel subtypes of crustin I. Consequential phylogenetic analysis provides the basis for reconstruction of possible evolutionary history of this group, which  is of special interest. A set of standard and widely used methods were employed   to obtain solid and rigorous data that well support the conclusions. Gene expression analysis showed  that several crustin genes were significantly up-regulated during molting stage as well as when exposed to infections. Notably, one of the genes is highly expressed in ovary and during early development stages. This work expands our knowledge of structural organization of crustins – a large family of invertebrate AMPs. The authors’ contributions may be important in future research of crustin biological functions and their evolutionary relationships. The paper is clear, concise and very well written.

Although I found the above merit in the authors’ findings, I identified some points that must be addressed before the work can be published in Marine Drugs.  

  1. There is no doubt that any crustacean uses a set of different AMPs - representatives of various structural families. The question arises why the host needs 34 AMPs which are similar in their function. Moreover, half of them are colocalized in the epidermis. It would be advisable that the authors briefly discuss this matter in comparison with other animals that express repertoires of homologous AMPs.
  2. It is a bit surprising that all 34 genes are functionally active. Have the gene regulatory regions been analyzed? Have the crustin pseudogenes been identified?
  3. It would be interesting if  the authors  make an assumption why only 15 out of 34 crustins were found at the genome level. It is also necessary to add reference (Lai and Aboobaker, BMC Genomics  (2017) 18:389. DOI 10.1186/s12864-017-3769-4) where the vannamei genome has already been analyzed and 15 genes encoding crustins have been identified.
  4. Discussion of possible evolution of the vannamei crustin genes is of special interest. Despite being rather speculative, a proposed model (Fig. 10) gives a reasonable explanation of crustin evolution. However, the assumption that “Type Ib crustins should come from Type Ia crustins, through a mutation of the stop codon generating a longer C-terminal region” has to be supported by a particular case with gene analysis, especially 3′UTRs.
  5. Data on participation of different crustins in vannamei molting process (Fig. 9) have been already described in the previous study (Nat Commun. 2019 Jan 21;10(1):356. doi: 10.1038/s41467-018-08197-4). If these data are the same, it must be noted in the text.
  6. Lines 325-326: please, provide quantitative criteria of crustin or crustin-like sequences identification.

Minor:

  1. Line 44: replace “AMPs” with “AMP”
  2. Line 71: replace “anti-bacteria” with “antibacterial”
  3. Line 85: - fill the gap “PP-Cru from … could”
  4. Figure 4: add A and B  in the figure legend
  5. Figure 4: add a,b,c,d,  in the figure legend
  6. Line 168: replace “brand” with “branch”

Author Response

  1. There is no doubt that any crustacean uses a set of different AMPs - representatives of various structural families. The question arises why the host needs 34 AMPs which are similar in their function. Moreover, half of them are colocalized in the epidermis. It would be advisable that the authors briefly discuss this matter in comparison with other animals that express repertoires of homologous AMPs.

Response: Thanks for the suggestion. We think that the following possible reasons might explain why so many Crustins exist and half of them are colocalized in the epidermis.

For one reason, different AMPs combinations might form defence cocktails to reduce the risk of infection [1]. For another reason, although AMPs have been always regarded as immune effectors which show wide activities against different microbes, more and more studies reveal that some AMPs also exhibit exquisite specificity against certain pathogens [see review article 2]. Therefore, a large number of AMPs are essential for the host’s survival in the marine environment which contains various microorganisms including pathogenic bacteria and viruses.

In crustaceans, crustins are regarded as the important immune effectors which participate in the first-line of host defense to combat any invaders [3]. High expression in epidermis, which directly contacts with seawater, will facilitate their immune defense functions. Furthermore, we also discussed that these Crustins might have functions in calcium deposition in shrimp during molting, as other WAP domain-containing proteins like Perlwapin in Haliotis laevigata shows similar function [4].

The above discussions have been added in the revised manuscript.

References:

(1) Zanchi, C.; Johnston, P.R.; Rolff, J. Evolution of defence cocktails: Antimicrobial peptide combinations reduce mortality and persistent infection. Mol Ecol 2017, 26(19), 5334-5343, doi:10.1111/mec.14267.

(2) Hanson, M.A.; Lemaitre, B. New insights on Drosophila antimicrobial peptide function in host defense and beyond. Curr Opin Immunol 2020, 62, 22-30, doi:10.1016/j.coi.2019.11.008.

(3) Jiravanichpaisal, P.; Narongsak, P.; Petkon, S.; Donnuea, S.; Soderhall, I.; Soderhall, K. Expression of immune-related genes in larval stages of the giant tiger shrimp, Penaeus monodon. Fish Shellfish Immun 2007, 23, 815-824, doi:10.1016/j.fsi.2007.03.003.

(4) Treccani, L.; Mann, K.; Heinemann, F.; Fritz, M. Perlwapin, an abalone nacre protein with three four-disulfide core (whey acidic protein) domains, inhibits the growth of calcium carbonate crystals. Biophys J 2006, 91, 2601-2608, doi:10.1529/biophysj.106.086108.

  1. It is a bit surprising that all 34 genes are functionally active. Have the gene regulatory regions been analyzed? Have the crustin pseudogenes been identified?

Response: Although these genes haven’t been detected in their protein level, all the 34 genes can be found in the transcriptome database, which indicates that these genes are all functionally active. The gene regulatory regions were not analyzed. We will analyze the information in the future study. Actually, more than 40 genes were identified after blast using WAP domain as query sequence. However, some sequences are not complete and can not be found in the transcriptome. We were not sure whether these sequences were crustins, other WAP-containing genes or crustin pseudogenes. Therefore, these sequences were not used in the study.

  1. It would be interesting if the authors make an assumption why only 15 out of 34 crustins were found at the genome level. It is also necessary to add reference (Lai and Aboobaker, BMC Genomics (2017) 18:389. DOI 10.1186/s12864-017-3769-4) where the vannamei genome has already been analyzed and 15 genes encoding crustins have been identified.

Response: In the Results 2.1, the sentence “Fifteen of them (including three previously characterized sequences) have been predicted in our previous shrimp genome database…” might be a bit confusing. These 15 crustins were annotated in the genome database. Actually, 33 out of 34 crustins (except LvCruIII-4) could be mapped to the genome database, whereas some of them were incomplete or without annotation based on the current version of the genome database. We have modified “predicted” into “annotated” in the sentence and added the information in Result 2.1 in the revised manuscript. The results of re-mapping of crustin genes to the shrimp genome database were added as a supplementary data (Figure S1) in the revised manuscript. The reference (Lai and Aboobaker, BMC Genomics (2017) 18:389. DOI 10.1186/s12864-017-3769-4), which identified 16 crustins from a L. vannamei transcriptome, was also added in the revised manuscript.

  1. Discussion of possible evolution of the vannamei crustin genes is of special interest. Despite being rather speculative, a proposed model (Fig. 10) gives a reasonable explanation of crustin evolution. However, the assumption that “Type Ib crustins should come from Type Ia crustins, through a mutation of the stop codon generating a longer C-terminal region” has to be supported by a particular case with gene analysis, especially 3′UTRs.

Response: Thanks for the constructive suggestion. We analyzed the nucleotide sequences of Type Ia Crustins and found that mutation of the stop codon in some sequences would lead to an extension of the C-terminal length. We have added the information following the assumption in the Discussion part of the revised manuscript.

  1. Data on participation of different crustins in vannamei molting process (Fig. 9) have been already described in the previous study (Nat Commun. 2019 Jan 21;10(1):356. doi: 10.1038/s41467-018-08197-4). If these data are the same, it must be noted in the text.

Response: Thanks for the reminder. Most of the expression data of the ten crustins in L. vannamei molting process have been described in our previous study. In order to keep the originality of the present data, we would like to remove the related small part of result. We have cited and discussed this content in the revised manuscript. We think that this would not affect the integrity of the article.

  1. Lines 325-326: please, provide quantitative criteria of crustin or crustin-like sequences identification.

Response: The sequences which hit a reported Crustin or Crustin-like sequence with the E-value less than 1E-5 were regarded as Crustins in shrimp. We have added the quantitative criteria in the revised manuscript.

Minor:

  1. Line 44: replace “AMPs” with “AMP”
  2. Line 71: replace “anti-bacteria” with “antibacterial”
  3. Line 85: - fill the gap “PP-Cru from … could”
  4. Figure 4: add A and B in the figure legend
  5. Figure 4: add a,b,c,d, in the figure legend
  6. Line 168: replace “brand” with “branch”

Response: Thanks for the comment. The mistakes have been modified in the revised manuscript.

Reviewer 2 Report

Marine Drugs

This manuscript reports a study of different sequences of crustin antimicrobial peptides from the shrimp Litopenaeus vannamei. The authors present results of a screening for crustin encoding sequences in L. vannamei in databases, and here they refer to the published genome of this species (one male)  in 2019 Nature Communication (https://doi.org/10.1038/s41467-018-08197-4).  However, they also used RNA-seq data from L. vannamei at different developmental stages published 2014 (most likely several individuals). The manuscript does not

The terminology in the manuscript is not perfectly clear to this reviewer. In the abstract the authors state that they “identified 34 full-length crustin encoding genes”, and in the discussion they conclude that “Existence of multiple crustin genes in one species provides a basis to study the evolution of this kind of genes in crustacean”. However, they base these conclusions not only on the genome database (one individual) but also on transcript sequences (several individuals). It is not clear from this paper (and not in https://doi.org/10.1038/s41467-018-08197-4) exactly how many crustin genes there are in the L. vannameigenome, or if several transcripts are the result of i) allelic variation, ii) alternative splicing. In order to draw conclusions, the full gene sequence (including intron/exon structure) of each crustin gene has to be clarified. In the genome paper  (https://doi.org/10.1038/s41467-018-08197-4), thirteen crustin genes are reported (although the gene structures are not shown). In order to state the number of genes present the authors need to give proof of the gene structure.

Another problem with this manuscript is the phylogenetic analysis, which this reviewer is not valid due to extremely low support values (only four nodes have values over 70, and the basal node value is 21 and can’t be considered supportive), i.e. the trees are not at all reliable and don’t show any phylogenic or evolutionary relationship.

For the expression studies, no statistic evaluation or methods are described.

In summary, in this reviewers opinion this manuscript is not suitable for publication, mainly due to severe methodological problems.

Specific comments:

Table 1: the sequences could not be detected by the NCBI Accession numbers given. When searching for XP_027238377 several transcripts hits appear with other numbers (GenBank: GFRP01024607.1, GGQV01084025, GHXV01177407.1 etc.).

Figures 4-5: The phylogenetic analysis presented here have very low (if any) support, and therefore the conclusions about evolutionary relationship as for example presented in figure 10 is very speculative.

Figure 6-7 and 9: No statistics is presented for the data in figures 6-7. How many replicates were used, and what about individual variation in the sequences.

Figure 8: In this figure the expression in the heatmap represent the mean of three replicates of three pooled samples, but nothing is said about the variation, and how the significance was calculated. The expression of different crustins after Vibrio and WSSV challenge has been covered in a high number of published papers (PubMed hits Crustin+Vibrio= 76; Crustin+WSSV=48). Are the results in this paper different from other studies? Please compare.

Author Response

  1. This manuscript reports a study of different sequences of crustin antimicrobial peptides from the shrimp Litopenaeus vannamei. The authors present results of a screening for crustin encoding sequences in L. vannamei in databases, and here they refer to the published genome of this species (one male) in 2019 Nature Communication (https://doi.org/10.1038/s41467-018-08197-4). However, they also used RNA-seq data from L. vannamei at different developmental stages published 2014 (most likely several individuals). The manuscript does not

Response: The comment was not complete.

  1. The terminology in the manuscript is not perfectly clear to this reviewer. In the abstract the authors state that they “identified 34 full-length crustin encoding genes”, and in the discussion they conclude that “Existence of multiple crustin genes in one species provides a basis to study the evolution of this kind of genes in crustacean”. However, they base these conclusions not only on the genome database (one individual) but also on transcript sequences (several individuals). It is not clear from this paper (and not in https://doi.org/10.1038/s41467-018-08197-4) exactly how many crustin genes there are in the L. vannamei genome, or if several transcripts are the result of i) allelic variation, ii) alternative splicing. In order to draw conclusions, the full gene sequence (including intron/exon structure) of each crustin gene has to be clarified. In the genome paper (https://doi.org/10.1038/s41467-018-08197-4), thirteen crustin genes are reported (although the gene structures are not shown). In order to state the number of genes present the authors need to give proof of the gene structure.

Response: Thanks for the suggestion. We re-mapped the nucleotide sequences of the 34 identified crustin genes to the genome database and found that 33 of them (except LvCruIII-4) could be mapped to the genome database, whereas some of them were incomplete based on the current version of genome database. We considered that this might be because the present genome database was not well-assembled enough. However, the results support the identification of these crustin genes in one species. We have added the related methodology (Materials and Methods 4.2) and results (Supplementary Figure S1) in the revised manuscript.

  1. Another problem with this manuscript is the phylogenetic analysis, which this reviewer is not valid due to extremely low support values (only four nodes have values over 70, and the basal node value is 21 and can’t be considered supportive), i.e. the trees are not at all reliable and don’t show any phylogenic or evolutionary relationship.

Response: The phylogenetic analysis was performed again and a tree with high support for the branches (Figure 5) was redrawn in the revised manuscript.

  1. For the expression studies, no statistic evaluation or methods are described.

Response: Thanks for the suggestion. The description of statistical analysis method for the immune challenge data (Figure 8) has been added in Materials and Methods 4.6 of the revised manuscript. The statistical analysis result was also provided as a supplementary Table (Table S1) in the revised manuscript.

The data in Figure 6 and 7 is based on the mixed sample of each tissue from 10 shrimps or in each development stage of many individuals (more than 500 individuals for embryonic stages and nauplii stages and about 30 individuals for other stages). The expression data show a mean expression level of the genes in each sample. Therefore, we did not add statistic analysis method for these parts.

The data in Figure 9 is based on three replicates. However, as most of this data have been described in our previous study (Nat Commun. 2019 Jan 21;10(1):356. doi: 10.1038/s41467-018-08197-4), we would like to remove the related small part of result to keep the originality of the present data. We have cited and discussed this content in the revised manuscript. We think that this would not affect the integrity of the article.

  1. In summary, in this reviewer’s opinion this manuscript is not suitable for publication, mainly due to severe methodological problems.

Response: The methodology with problems has been carefully modified according to the reviewer’s suggestions in the revised manuscript.

Specific comments:

  1. Table 1: the sequences could not be detected by the NCBI Accession numbers given. When searching for XP_027238377 several transcripts hits appear with other numbers (GenBank: GFRP01024607.1, GGQV01084025, GHXV01177407.1 etc.).

Response: We have checked again and confirmed that all the cited accession numbers could be found in NCBI website. One could search in the NCBI homepage with these accession numbers, selecting the “All Database” for search. The accession numbers of the newly submitted sequences in the present study will be release upon their appearance in any publication.

  1. Figures 4-5: The phylogenetic analysis presented here have very low (if any) support, and therefore the conclusions about evolutionary relationship as for example presented in figure 10 is very speculative.

Response: Thanks for the suggestion. The phylogenetic analysis was performed again and a tree with high support for the branches was redrawn in the revised manuscript.

  1. Figure 6-7 and 9: No statistics is presented for the data in figures 6-7. How many replicates were used, and what about individual variation in the sequences.

Response: The data in Figure 6 and 7 is based on the mixed sample of each tissue from 10 shrimps or in each development stage of many individuals (more than 500 individuals for embryonic stages and nauplii stages and about 30 individuals for other stages). The expression data show a mean expression level of the genes in each sample.

The shrimps for tissue collection were cultured for a long time in our laboratory. Through manipulating the water quality and culture condition, the shrimps were in good health. Individuals with similar body weight and body length were used for sampling. It would greatly reduce individual differences and could assure the reliability of the expression trends of tested genes. The samples of different developmental stages were collected from the offspring of three shrimp and were also cultured in our laboratory.

The data in Figure 9 is based on three replicates. However, as most of this data have been described in our previous study (Nat Commun. 2019 Jan 21;10(1):356. doi: 10.1038/s41467-018-08197-4), we would like to remove the related small part of result to keep the originality of the present data. We have cited and discussed this content in the revised manuscript. We think that this would not affect the integrity of the article.

  1. Figure 8: In this figure the expression in the heatmap represent the mean of three replicates of three pooled samples, but nothing is said about the variation, and how the significance was calculated. The expression of different crustins after Vibrio and WSSV challenge has been covered in a high number of published papers (PubMed hits Crustin+Vibrio= 76; Crustin+WSSV=48). Are the results in this paper different from other studies? Please compare.

Response: Thanks for the suggestion. The description of statistical analysis method for the immune challenge data (Figure 8) has been added in Materials and Methods 4.6 of the revised manuscript. The statistical analysis result was also provided as a supplementary Table (Table S1) in the revised manuscript. The results were also discussed with published literatures in the revised manuscript.

Reviewer 3 Report

This paper articulates the identification and functional diversity of crustins within Litopenaeus vannamei. Furthermore, it goes someway to outline the evolutionary origin of these peptides. Although an overall interesting article there are issues which hinder its publication in its current form.

  1. General spelling and grammar throughout should be addressed. The attached annotated document outlines these.

The results section of the report is well presented in the main however, there are some issues to be addressed.

  1. Line 118, the text reads “whereas other three crustins” in the sentence this isn’t clear what the authors mean, is it the remaining LvCrustin III peptides? This should be reworded to be clearer.
  2. Figure 2 and Figure 3 lack quality. They are poor images and are not readable in their current form. You should consider redrawing these images, highlighting better the regions which you are intending to discuss and make much clearer the other information included (pI/Mw and Types)
  3. Figure 5 should be labelled more clearly. The branches labelled “a-d” are lost in the figure and the text so suggest the authors capitalise these.
  4. Consider Figure 6 and Figure 7 redraw as one is spread over two pages, the headings to the columns could be improved and the highlighting of the specific crustins mentioned in the text could be made as this is difficult to discern from the current figures.
  5. Figure 9 in the text mentions A and B yet the figure itself has no A or B, please revise this.

Whilst the discussion leads the reader nicely through the analysis of the genomic data, there are issues here.

  1. Line 234, the sentence here is contradictory. The authors should consider the sentence inclusion and if this can be clarified, it requires a reference.
  2. Line 239, the sentence reads “Plenty of protein-coding genes and polymorphic sites might enable shrimp to defend against different pathogens” The authors need to provide evidence for this and use of the word “plenty” is colloquial, it has no basis to deduce a significance.

Author Response

  1. General spelling and grammar throughout should be addressed. The attached annotated document outlines these.

Response: Thanks for your revision of the spelling and grammar. We have carefully revised the manuscript again.

The results section of the report is well presented in the main however, there are some issues to be addressed.

  1. Line 118, the text reads “whereas other three crustins” in the sentence this isn’t clear what the authors mean, is it the remaining LvCrustin III peptides? This should be reworded to be clearer.

Response: The phrase“whereas other three crustins” referred to the remaining LvCrustin III peptides. We modified the description in the revised manuscript.

  1. Figure 2 and Figure 3 lack quality. They are poor images and are not readable in their current form. You should consider redrawing these images, highlighting better the regions which you are intending to discuss and make much clearer the other information included (pI/Mw and Types)

Response: High quality images of these Figures have been used in the revised manuscript.

  1. Figure 5 should be labelled more clearly. The branches labelled “a-d” are lost in the figure and the text so suggest the authors capitalize these.

Response: Thanks for the suggestion. We have used capitalized letters in Figure 5 of the revised manuscript.

  1. Consider Figure 6 and Figure 7 redraw as one is spread over two pages, the headings to the columns could be improved and the highlighting of the specific crustins mentioned in the text could be made as this is difficult to discern from the current figures.

Response: Thanks for the suggestion. We have redrawn Figure 6 and Figure 7 to make the information clearer.

  1. Figure 9 in the text mentions A and B yet the figure itself has no A or B, please revise this.

Response: Thanks for the reminder. The present Figure 9 showed Figure 9B. Figure 9A displayed the tissue distribution result of these Crustins. As most of this data have been described in our previous study (Nat Commun. 2019 Jan 21;10(1):356. doi: 10.1038/s41467-018-08197-4), we would like to remove the related small part of result to keep the originality of the present data. We have cited and discussed this content in the revised manuscript. We think that this would not affect the integrity of the article.

Whilst the discussion leads the reader nicely through the analysis of the genomic data, there are issues here.

  1. Line 234, the sentence here is contradictory. The authors should consider the sentence inclusion and if this can be clarified, it requires a reference.

Response: The sentence has been modified and two related references have been cited in the revised manuscript.

  1. Line 239, the sentence reads “Plenty of protein-coding genes and polymorphic sites might enable shrimp to defend against different pathogens” The authors need to provide evidence for this and use of the word “plenty” is colloquial, it has no basis to deduce a significance.

Response: We have modified the sentence into “We infer that a large number of protein-coding genes are essential for the shrimp to defend against different pathogens in the marine environment” in the revised manuscript.

Reviewer 4 Report

At its core, this manuscript provides a careful annotation of the crustin genes found in the sequenced genome of one species of shrimp. A (somewhat problematic) phylogenetic analysis is included. Also included is information relevant to inferring function, including expression profiles across tissues, across ontogenetic stages, and in response to an immune-system challenge.  There is not quite enough here to make any very strong inferences about either function or evolution, but the article nonetheless provides useful background for any subsequent functional or evolutionary study.

This is an essentially descriptive paper, which provides a rich annotation for one class of genes in one organism.  It is not a high-profile paper but probably merits publication in some venue -- either in a specialized journal such as Marine Drugs, or in a determinedly nonselective journal such as PLoS One or Scientific Reports.

There are deficiencies in the presentation of the phylogenetic results, which should be addressed, as follows.

Figure 4 caption and Figure 5 caption.  There are a few points of confusion here regarding the bootstrap analysis.  (1) Say in the caption what the numbers below the internodes mean.  My guess is that they are bootstrap percentages (also called "bootstrap support values" or "bootstrap values"), but this needs to be stated explicitly. (2) "Bootstrap value was set at 1000" is not a correct statement.  Presumably this refers to the number of bootstrap replicates performed.  (3) It is strange to see such low bootstrap percentages on so many nodes.  Usually when one performs a bootstrap analysis, the output is a majority-rule consensus tree, such that only nodes with values above 50% are displayed.  Here it looks like the authors are displaying the single NJ tree with bootstrap percentages (from a separate bootstrap analysis) transcribed onto it.  This should stated explicitly. 

A bigger issue that I have some qualms about: 

Usually researchers who perform a bootstrap analysis do so in order to assess the degree of confidence merited by the phylogenetic results.  Here the authors perform the bootstrap analysis and (apparently) give its results, but then entirely ignore those results and their implications  in the discussion.  If the bootstrap results are taken seriously, most of the structure of the trees (Figs 4 & 5) goes away.  The authors say that the analysis shown in Figure 4 divides the genes into 2 large groups A and B, but one of those groups (A) is paraphyletic and the node subtending the monophyletic group B apparently has extremely low bootstrap support (25%); it's not really clear why the authors discern 2 groups (as opposed to more than 2) or why they divide them where they do.  I do not think that this is necessarily a make-or-break issue. The authors can make the case that the their NJ analysis provides a reasonable starting point for organizing the diversity of this gene family, but if they are going to present bootstrap results they should acknowledge that these values are very low and that the analysis that they present is highly tentative.

            A related issue is that the authors could have done a great deal more to find good trees.  Alignment by eye (or clustal alignment followed by adjustment by eye, which amounts to the same thing), followed by a neighbor-joining analysis, is often sufficient to generate a plausible phylogenetic hypothesis.  But the rock-bottom bootstrap values shown here (if that is what they are) suggest that a different approach is probably warranted in this case.  The ragged alignment implies that alignment is a significant problem.  The authors might try a more sophisticated approach to simultaneous alignment and phylogeny estimation, such as PASTA (Mirarab, S., Nguyen, N., Guo, S., Wang, L.-S., Kim, J. & Warnow, T. (2015) PASTA: Ultra-large multiple sequence alignment for nucleotide and amino-acid sequences. Journal of Computational Biology 22, 377–386).  I do not insist on this, as I think that article's value derives more from its gene annotations than from the exact topology of the trees.  But unless better trees are sought, the authors do need to adopt a more circumspect tone in discussing the phylogenetic results.

Additional minor comments and copy-edits:

line 42, change "to different microbial containing environments" to "to microbial challenges in different environments".

line 89, change "play multi-functionalities" to "play diverse roles".

lines 97-99 (final sentence of introduction) change from past tense to present tense.

Figures 2 and 3 are of poor quality and illegible (or barely legible and pixelated when electronically enlarged). Please improve.

line 228 change "previous identified crustins" to "previously identified crustins"

line 233-234, delete the sentence "However, the present identified crustins. . . " (This information is conveyed earlier, in line 229.)

line 242-244, "Therefore, we infer that a large number of Crustin encoding genes. . . " This is a nonsequitur.  It's a reasonable conjecture, but it doesn't follow from the premises previously given; therefore the "Therefore" is inappopriate.

line 252-253, meaning of "the same or close evolutionary origins" is unclear.  This whole section should be re-worked to acknowledge the tentativeness of the phylogenetic analysis. And probably shortened accordingly.

Throughout: be consistent about whether "crustin" is capitalized or not.  Probably it should not be.

line 287, change "more solid evidence is" to "comparative studies with broad taxonomic sampling and rigorous phylogenetic analysis are"

Figure 9, change "mutation" to "substitution".  Insertions and deletions arise from mutations also.

Author Response

1. There are deficiencies in the presentation of the phylogenetic results, which should be addressed, as follows. Figure 4 caption and Figure 5 caption. There are a few points of confusion here regarding the bootstrap analysis. (1) Say in the caption what the numbers below the internodes mean. My guess is that they are bootstrap percentages (also called "bootstrap support values" or "bootstrap values"), but this needs to be stated explicitly. (2) "Bootstrap value was set at 1000" is not a correct statement.  Presumably this refers to the number of bootstrap replicates performed. (3) It is strange to see such low bootstrap percentages on so many nodes. Usually when one performs a bootstrap analysis, the output is a majority-rule consensus tree, such that only nodes with values above 50% are displayed.  Here it looks like the authors are displaying the single NJ tree with bootstrap percentages (from a separate bootstrap analysis) transcribed onto it. This should be stated explicitly. A bigger issue that I have some qualms about: Usually researchers who perform a bootstrap analysis do so in order to assess the degree of confidence merited by the phylogenetic results. Here the authors perform the bootstrap analysis and (apparently) give its results, but then entirely ignore those results and their implications in the discussion. If the bootstrap results are taken seriously, most of the structure of the trees (Figs 4 & 5) goes away. The authors say that the analysis shown in Figure 4 divides the genes into 2 large groups A and B, but one of those groups (A) is paraphyletic and the node subtending the monophyletic group B apparently has extremely low bootstrap support (25%); it's not really clear why the authors discern 2 groups (as opposed to more than 2) or why they divide them where they do. I do not think that this is necessarily a make-or-break issue. The authors can make the case that the NJ analysis provides a reasonable starting point for organizing the diversity of this gene family, but if they are going to present bootstrap results, they should acknowledge that these values are very low and that the analysis that they present is highly tentative.

Response: Thanks for the comment. The bootstrap values described in Figure 4 and Figure 5 captions meant bootstrap percentages. The sentence “Bootstrap value was set at 1000” has been changed into “Bootstrap replicates were set at 1000” in the revised manuscript. In figure 4, many bootstrap percentages were low since the WAP sequences of all identified crustins in L. vannamei were used for analysis. As this figure (Figure 4) was to exhibit the relationship of all identified crustins in the species, we redrew the figure without setting bootstrap replicates. The corresponding interpretation of this figure in the first paragraph of Result 2.3 has been revised. We have also considered the comment that “it’s not clear to discern two groups” in Figure 4 and revised the description on it. In the new Figure 4, different types of crustins were classified into different groups. Except Ia-2, Type I crustins exhibited close relationship with each other. Type III and IV crustins were also showed close relationship with Type I crustins. Type II crustins were all clustered into one big group, together with Ia-2, Type VI and VII crustins. The topology of these crustins was consistent with previous data.

2. A related issue is that the authors could have done a great deal more to find good trees. Alignment by eye (or clustal alignment followed by adjustment by eye, which amounts to the same thing), followed by a neighbor-joining analysis, is often sufficient to generate a plausible phylogenetic hypothesis. But the rock-bottom bootstrap values shown here (if that is what they are) suggest that a different approach is probably warranted in this case. The ragged alignment implies that alignment is a significant problem. The authors might try a more sophisticated approach to simultaneous alignment and phylogeny estimation, such as PASTA (Mirarab, S., Nguyen, N., Guo, S., Wang, L.-S., Kim, J. & Warnow, T. (2015) PASTA: Ultra-large multiple sequence alignment for nucleotide and amino-acid sequences. Journal of Computational Biology 22, 377–386). I do not insist on this, as I think that article's value derives more from its gene annotations than from the exact topology of the trees. But unless better trees are sought, the authors do need to adopt a more circumspect tone in discussing the phylogenetic results.

Response: Thanks for the suggestions. Actually, we have tried several methods and used different sets of WAP sequences to perform the phylogenetic analysis. The present phylogenetic tree was the best result that we have ever tried. Thanks for the reviewer’s understanding on the imperfect phylogenetic tree. To express the data in a more rigorous manner, we have modified the related discussions into a discreet tone in the revised manuscript.

3. Additional minor comments and copy-edits:

line 42, change "to different microbial containing environments" to "to microbial challenges in different environments".

Response: The sentence has been improved in the revised manuscript.

line 89, change "play multi-functionalities" to "play diverse roles".

Response: The word has been replaced in the revised manuscript.

lines 97-99 (final sentence of introduction) change from past tense to present tense.

Response: The tense has been changed in the revised manuscript.

Figures 2 and 3 are of poor quality and illegible (or barely legible and pixelated when electronically enlarged). Please improve.

Response: Images with high resolution have been submitted in the revised manuscript.

line 228 change "previous identified crustins" to "previously identified crustins"

Response: The phrase has been replaced in the revised manuscript.

line 233-234, delete the sentence "However, the present identified crustins. . . " (This information is conveyed earlier, in line 229.)

Response: The second half of the sentence, “…which was previously identified only in some ant genomes”, has been deleted as it was conveyed earlier. The first part of the sentence, “However, the present identified crustins don’t contain Type V crustin either”, expresses that L. vannamei does not encode Type V crustin although so many crustin genes are in the genome. Therefore, the content was not deleted in the revised manuscript.

line 242-244, "Therefore, we infer that a large number of Crustin encoding genes. . . " This is a nonsequitur. It's a reasonable conjecture, but it doesn't follow from the premises previously given; therefore the "Therefore" is inappropriate.

Response: The sentence has been changed into “We guess that a large number of Crustin encoding genes…” in the revised manuscript.

line 252-253, meaning of "the same or close evolutionary origins" is unclear. This whole section should be re-worked to acknowledge the tentativeness of the phylogenetic analysis. And probably shortened accordingly.

Response: The section of phylogenetic discussion has been shortened and re-written in a circumspect tone in the revised manuscript.

Throughout: be consistent about whether "crustin" is capitalized or not. Probably it should not be.

Response: The word “crustin” has been used in the revised manuscript.

line 287, change "more solid evidence is" to "comparative studies with broad taxonomic sampling and rigorous phylogenetic analysis are"

Response: Thanks for the advice. The sentence has been modified in the revised manuscript.

Figure 9, change "mutation" to "substitution". Insertions and deletions arise from mutations also.

Response: The word "mutation" has been changed to "substitution" in Figure 9 and the main text.

Round 2

Reviewer 2 Report

The manuscript “Molecular and functional diversity of crustin genes provides insight into their evolution in the shrimp Litopenaeus vannamei” has been submitted in a revised version. The manuscript authors state that they have screened the genome of Litopenaeus vannamei for crustin sequences and could detect 34 crustin encoding genes. They have used known transcript sequences from mixed RNA samples (i.e. several individuals) and done BLAST search against the genome (http://www.shrimpbase.net/vannamei.html). Unfortunately this database is not reachable for this reviewer, but genome information can also be found at (https://www.ncbi.nlm.nih.gov/genome/10710).

The paper is very confusing, and most of all regarding the conclusions about evolution and phylogeny. The new supplementary figure S1 doesn’t give any understandable information about gene sequences. No information about Scaffold ID, Locus #, Start and stop codon, Gene ID is given.

The basis for categorization of transcripts into the different “Gene symbol” (in Table 1) is not clear. What is common to everyone with for example "Gene symbol = Ia" appears to be the fact that they have a signal peptide, a cysteine-rich domain and a WAP domain. However, not all proteins with such a structure are crustins. The transcripts with Gene symbol = Ia-4 and Ia-10, looks as if they encode proteins that are more similar to perlwapin or papilin-like proteins. These are probably protease inhibitors, and many protease inhibitors also contain the WAP domains. The phylogenetic analysis is still not valid, and it is clear from figure 2 (very low quality), as well as figure 3-5 that no information or conclusions can be drawn from these (low support values and several polytomies indicating of lack of reliable phylogenetic information (fig 5)).

The main information in this manuscript is: There are several (34) different transcripts (and perhaps genes) encoding secreted proteins with WAP-domains, together with various cysteine-rich, glycine-rich, serine/leucine-rich etc. regions in different combinations. A few of these have been shown to have crustin-like antimicrobial activity, (but most of them have unknown function and can possible be protease inhibitors or have other functions). The expression of these 34 transcripts differ between tissues (according to figure 6) and during development (according to figure 7), and some change their expression after infection with Vibrio parahaemolyticus or WSSV.

Therefore, my suggestion is that the authors rewrite their manuscript, change the title, and remove everything about evolution and phylogeny (figures 2,3,4, and 5), and keep figures 1,6,7 an 8. Of course the name crustin should be changed to WAP-domain containing proteins.

Author Response

  1. The manuscript “Molecular and functional diversity of crustin genes provides insight into their evolution in the shrimp Litopenaeus vannamei” has been submitted in a revised version. The manuscript authors state that they have screened the genome of Litopenaeus vannamei for crustin sequences and could detect 34 crustin encoding genes. They have used known transcript sequences from mixed RNA samples (i.e. several individuals) and done BLAST search against the genome (http://www.shrimpbase.net/vannamei.html). Unfortunately this database is not reachable for this reviewer, but genome information can also be found at (https://www.ncbi.nlm.nih.gov/genome/10710).

Response: The website of the online shrimp genome database maintained by our lab is public. We are sorry to hear that it is not reachable for the reviewer. We will check it soon. However, we have done the BLAST search against the genome database at NCBI and updated the result in the revised manuscript.

  1. The paper is very confusing, and most of all regarding the conclusions about evolution and phylogeny. The new supplementary figure S1 doesn’t give any understandable information about gene sequences. No information about Scaffold ID, Locus #, Start and stop codon, Gene ID is given.

Response: Thanks for the suggestion. We have provided detailed sequence information including scaffold ID, locus (Scaffold start and end sites), gene ID and the completeness of ORF for the BLAST result. In order to show the information clearer, we have changed the supplementary Figure S1 with a supplementary Table S1.

As for the conclusion of evolution and phylogeny, it was mainly inferred from the results of phylogenetic analysis (Figure 5) and the sequence features of different types of Crustins (Figure 1 and 2). We are thankful to the reviewer for giving constructive suggestion that the Figure 5 should be improved. We have redrawn it again and generated a binary tree, which provided support to the supposition of Crustin evolution.

  1. The basis for categorization of transcripts into the different “Gene symbol” (in Table 1) is not clear. What is common to everyone with for example "Gene symbol = Ia" appears to be the fact that they have a signal peptide, a cysteine-rich domain and a WAP domain. However, not all proteins with such a structure are crustins. The transcripts with Gene symbol = Ia-4 and Ia-10, looks as if they encode proteins that are more similar to perlwapin or papilin-like proteins. These are probably protease inhibitors, and many protease inhibitors also contain the WAP domains. The phylogenetic analysis is still not valid, and it is clear from figure 2 (very low quality), as well as figure 3-5 that no information or conclusions can be drawn from these (low support values and several polytomies indicating of lack of reliable phylogenetic information (fig 5)).

Response: Crustins are defined as one kind of secretory antimicrobial peptides that contain one or two WAP domain (s) (see review articles 1 and 2). In crustaceans, proteins with such a sequence feature are considered as crustins. However, proteins with signal peptide and WAP domains have different names in different kinds of animals, such as the perlwapin in abalones, which has a signal peptide and three WAP domains (reference 3). In crustaceans, a recent study also reported a protein with three WAP domains in Procambarus clarkii, which was called triple WAP domain containing protein (reference 4). Crustins not only exhibited antibacterial activities, some of them also displayed proteinase inhibitory activities (reference 5). Therefore, the sequences identified in the present study were named as crustins.

We transposed Figure 2 and enlarged the image to make it clearer. An image with high quality was provided in the revised manuscript. The details of the image could be see clearly after amplifying it.

Figure 3 showed the identification of new Types of crustins in another shrimp Fenneropenaeus chinensis, which provided support of their identification in L. vannamei.

In Figure 4, we used all the WAP domain sequences from 34 identified Crustins in L. vannamei, which generated a tree with low support values. However, the figure mainly exhibited a classification of crustins in one species. The result did not express the evolutionary relationship among these sequences.

The conclusion of evolution and phylogeny, it was mainly inferred from the results of phylogenetic analysis (Figure 5) and the sequence features of different types of Crustins (Figure 1 and 2). The WAP domain sequences of different types crustins in different crustacean species were used to draw the figure (Figure 5), and WAP domains from Perlwapin in abalones were used as outgroup. We are thankful to the reviewer for giving constructive suggestion that the Figure 5 should be improved. We have redrawn it again and generated a binary tree, which provided support to the supposition of Crustin evolution.

References:

(1) Smith, V.J.; Fernandes, J.M.O.; Kemp, G.D.; Hauton, C. Crustins: Enigmatic WAP domain-containing antibacterial proteins from crustaceans. Dev Comp Immunol 2008, 32, 758-772.

(2) Tassanakajon, A.; Somboonwiwat, K.; Amparyup, P. Sequence diversity and evolution of antimicrobial peptides in invertebrates. Dev Comp Immunol 2015, 48, 324-341.

(3) Treccani, L.; Mann, K.; Heinemann, F.; Fritz, M. Perlwapin, an abalone nacre protein with three four disulfide core (whey acidic protein) domains, inhibits the growth of calcium carbonate crystals. Biophys J 2006, 91, 2601-2608.

(4) Zhang, Y.X.; Wang, J.X.; Wang, X.W. First identification and characterization of a triple WAP domain containing protein in Procambarus clarkii provides new insights into the classification and evolution of WAP proteins in crustacean. Fish Shellfish Immun 2019, 94, 592-598.

(5) Jia, Y.P.; Sun, Y.D.; Wang, Z.H.; Wang, Q.; Wang, X.W.; Zhao, X.F.; Wang, J.X. A single whey acidic protein domain (SWD)-containing peptide from fleshy prawn with antimicrobial and proteinase inhibitory activities. Aquaculture 2008, 284, 246-259.

  1. The main information in this manuscript is: There are several (34) different transcripts (and perhaps genes) encoding secreted proteins with WAP-domains, together with various cysteine-rich, glycine-rich, serine/leucine-rich etc. regions in different combinations. A few of these have been shown to have crustin-like antimicrobial activity, (but most of them have unknown function and can possible be protease inhibitors or have other functions). The expression of these 34 transcripts differ between tissues (according to figure 6) and during development (according to figure 7), and some change their expression after infection with Vibrio parahaemolyticus or WSSV. Therefore, my suggestion is that the authors rewrite their manuscript, change the title, and remove everything about evolution and phylogeny (figures 2,3,4, and 5), and keep figures 1,6,7 an 8. Of course the name crustin should be changed to WAP-domain containing proteins.

Response: As above-mentioned in the last response, Figure 2, 3, and 4 showed the sequence features of crustins in L. vannamei and the support of newly identified types in other species. Therefore, we would like to keep these figures in the manuscript. Although the conclusion of evolutionary analysis was not very solid, the conclusion was actually a supposition that tried to describe the relationship of different types crustins in crustacean. To make this point clear, we added the description “More evidence is needed to clarify the evolutionary relationship of crustin genes in crustaceans” in the end of third paragraph in Discussion, as well as in the Conclusion part. As explained in the last response, we would like to keep the name of crustins for these sequences.

Round 3

Reviewer 2 Report

The authors have clarified the sequences and how to find the genes in GenBank and the new Supplementary table S1 is very helpful. However, the manuscript is still entitled “Molecular and functional diversity of crustin genes provides insight into their evolution in the shrimp Litopenaeus vannamei”, and there are no changes made to certify the title. The phylogeny is not valid, and the naming of all WAP-domain proteins as crusting AMPs are still not verified by any experimental evidences. There are many WAP domain proteins. I can not recommend publication in its present form and again recommend the authors to change the focus and to rewrite their manuscript, change the title, and remove everything about evolution and phylogeny and change the name crustin should be changed to WAP-domain containing proteins.

Author Response

Response: Thanks for the comment. In figure 4, many bootstrap percentages were low since the WAP sequences of all identified crustins in L. vannamei were used for analysis. As this figure (Figure 4) was to exhibit the relationship of all identified crustins in the species, we redrew the figure without setting bootstrap replicates. The corresponding interpretation of this figure in the first paragraph of Result 2.3 has been revised. In the new Figure 4, different types of crustins were classified into different groups. Except Ia-2, Type I crustins exhibited close relationship with each other. Type III and IV crustins were also showed close relationship with Type I crustins. Type II crustins were all clustered into one big group, together with Ia-2, Type VI and VII crustins. The topology of these crustins was consistent with previous data.

Although the identified sequences have not been verified by functional experiments, the sequence features, including the signal peptide, cysteine-rich region, glycine-rich region and one or two WAP domains, of these genes show that they are crustin-like genes according to the definition of crustin genes in crustacean. To make it more reasonable, the name of these genes has been changed into “crustin-like genes” and their protein name has been changed into “putative crustins” in the revised manuscript.

We agreed with the reviewer’s opinion that the phylogenetic tree was imperfect. Actually, we have tried several methods and used different sets of WAP sequences to perform the phylogenetic analysis. The present phylogenetic tree was the best result that we have ever tried. Although the main content of the manuscript is to describe the identification, annotation and expression patterns of crustin-like genes, we would like to keep the phylogenetic analysis content since the data identified all of this kind of gene in one species and the data would provide an insight to think their evolutionary relationship, even though the it still needs to be improved. To express the data in a more rigorous manner, the discussion on the phylogenetic analysis has been shortened and rewritten in a circumspect tone in the revised manuscript. Hope these revisions could improve the manuscript to meet the requirement of the journal.